# Shortcomings of Modern Large-Scale Common Sense Knowledge Graphs⋆

Oleg Sychev[1][0000−0002−7296−2538]

Volgograd State Technical University, Lenin Ave, 28, Volgograd, 400005, Russia
oasychev@gmail.com

**Abstract.** In recent years, a significant number of large-scale common-sense knowledge graphs were developed using crowdsourcing and extracting data from natural-language texts. They are successfully used in several areas, but their usage was mostly restricted to information search and retrieval and increasing the accuracy of other methods. The ability to make conclusions using software reasoners over these knowledge graphs is limited: they contain more information than knowledge. In this work, I analyze these shortcomings and their causes, focusing on the types of knowledge that were missed during knowledge acquisition - the negative knowledge - and consider the ways to overcome these problems by enhancing common sense knowledge graphs. Drawing parallels with human intelligence, these enhanced graphs can be used in hybrid architectures with neural networks to develop trustworthy AI systems. The discussed concerns can be useful for automatically-constructed domain knowledge graphs as well.

**Keywords:** Common sense knowledge graphs · Hybrid reasoning.

## 1 Introduction

Recent advances in research and technology allowed building large-scale knowledge graphs in areas like scholarly data, programming, medicine. One of the actively developing research areas is common sense knowledge. Practically all recently developed major common-sense knowledge graphs – ConceptNet [3], WebChild [4], ATOMIC [2], and COMET [1] were developed using bottom-up approach. The main methods of acquiring information for them were crowdsourcing and information extraction.

However, despite the impressive sizes of these knowledge bases, their chief usage was limited to information retrieval and increasing the accuracy of the other methods. The amount of common sense reasoning is minimal. Often, when it comes to reasoning, researchers – even having access to the structured data – turn to using neural networks, embeddings, and frequencies instead of formal reasoning to make final conclusions. These methods give probabilistic, unstable results because of the high dimensionality of their input data and are often

---

⋆ The reported study was funded by RFBR, project number 20-07-00764.

vulnerable to adversarial attacks, making glaring errors that humans rarely do having much less knowledge [5].

Theoretically, formal, highly-structured data of knowledge graphs should be ideal for common sense reasoning. To understand why this area is stalling we should analyze the kinds of data represented in these knowledge graphs and their connection with human reasoning.

## 2    Positive and Negative Knowledge

Considering existing common sense knowledge graphs, it is easy to see that they can be used only under the Open World Assumption (OWA). The knowledge presented in them is not nearly complete to use the Closed World Assumption (CWA). For example, though the concept "dog" has 139 "CapableOf" links in ConceptNet, there is no connection to tell us that a dog is capable of sitting. It says that a dog can "win a blue ribbon", but nothing about winning ribbons of the other colors. WebChild contains only 5 activities with a dog, without even pet a dog or playing with a dog but containing eating a dog. According to ATOMIC, a person can call her dog because she wanted to feed the dog, but feeding is absent from the list of subsequent actions of this person; COMET says that a person may call her dog because she needed to get her phone.

We can define three kinds of knowledge using description logic. Consider role $R$, concept $C$ so that $\forall R.\,C$, and individual or concept $X$. A knowledge base contains *positive knowledge* about $X$ regarding $R$ if the information about individuals $c_+ \subset C$ so that $\forall c \in c_+ (x, c) \colon R$ can be queried from the base. A knowledge base contains *negative knowledge* about $X$ regarding $R$ if the information about individuals $c_- \subset C$ that are knowingly not linked by role $R$ to $X$ can be queried from the base. A knowledge base contains *full knowledge* about $X$ regarding $R$ if for any individual $c \in C$ the base can tell whether $(x, c) \colon R$ is true or false.

Under OWA, the knowledge that is not represented in the knowledge base is treated as unknown so the knowledge base has to explicitly define negative knowledge. However, the bulk of the information in existing knowledge graphs is devoted to positive knowledge. Among the many relations in ConceptNet, only relation "Desire" has its negative counterpart "NotDesires", while important relations like "CapableOf", "UsedFor" and "ReceivesAction" contain only positive knowledge. WebChild, ATOMIC, and COMET contain only positive knowledge.

However, negative knowledge plays an important role in human reasoning. While creativity (i.e. generation of new ideas and strategies) is often intuitive, it produces "harmonic" (according to Poincaré) but not necessarily correct solutions. Living beings need safe exploration strategies because the price of a mistake may be death. So one of the primary functions of human reasoning is verifying generated solutions and weeding out dangerous or obviously wrong ones. This is confirmed by the conservative nature of the human world-model. Assessing believability and pruning obviously wrong solutions is a characteristic feature of human reasoning, routinely used in tasks ranging from word sense

disambiguation, sentence parsing to question answering and analyzing the feasibility of a long-term strategy. In artificial intelligence, weeding out wrong options is a good strategy for common sense question answering to multiple-choice questions; it also can be used in natural language processing as natural-language syntax and semantics often rely on believability assessing for disambiguation. However, the negative knowledge required for it is mostly absent in existing common sense knowledge graphs. We know that a snake can slither (at least down the street) from ConceptNet be we do not know that a snake cannot walk.

Another important problem concerning common-sense knowledge graphs manifests itself when the subject of a relationship is a concept: what quantifier the link in the knowledge graph implies? It is inconclusive in ConceptNet: any dog is capable of drinking water and barking, but only some dogs are capable of pulling a sleigh of smelling drugs. ConceptNet does not give us any clue about the quantifier, forcing us to treat all the data as having an existential quantifier. The same applies to properties of concepts in WebChild and the relations ATOMIC. Basically, we get an unsystematic list of things at least one dog is capable of that is not quite useful in formal reasoning for most tasks.

## 3   Acquiring and Representing Negative Knowledge

To enhance common sense knowledge graphs we need to introduce more negative knowledge to them, balancing the dominating positive knowledge. However, this poses several problems regarding knowledge extraction and storage.

When extracting data from natural-language texts and encyclopedias, the vast majority of the information is positive knowledge. Negative knowledge is so implicit to our thinking that we rarely feel the need to write it down, much less to formalize it. So it is difficult to acquire by text mining. E.g. our textual sources do not contain explicit information that snakes cannot walk because it is too obvious for most people. It is not obvious for artificial intelligence, though. The algorithms of text mining may be enhanced to pay special attention to negative sentences they encounter, especially the negative statements about concepts (not individuals). The other important source of knowledge are sentences containing "only" and other forms of universal quantification: a statement that "snakes only slither and jump" contains full knowledge about possible movements by snakes.

The crowdsourcing approach is better suited for gathering negative knowledge if the crowdsourcing system would ask the correct questions. One good form of a crowdsourcing application to gather negative knowledge may be a game like "teach an alien about living on Earth" that allows gathering knowledge about what can and cannot be done and why.

Another problem is the amount of negative knowledge: the things an individual (or any individual belonging to a concept) cannot do are often more numerous than the things they can do. This can be solved by capturing full knowledge - i.e. drawing a precise border between true and false statements. OWL features like disjoint classes, covering and closure axioms can be used to capture full knowledge in a relatively small number of statements. Knowledge graphs also

should provide distinction between the universal and existential quantifiers when linking concepts.

For capturing specifically common sense knowledge, special techniques may be used. For example, a variation of universal quantifier meaning "the statement is true for any individual belonging to the concept except these for whom an exception is directly specified" will allow many negative-knowledge statements that are not exactly true but often true like "fish does not fly" that is generally true except Exocoetidae family.

## 4  Conclusion

During the construction of large common sense knowledge graphs, most of the attention was spent on capturing positive knowledge. However, this left out a large body of negative knowledge that is implicit in our thinking, while the incomplete information in these graphs does not allow using CWA. Negative knowledge is arguably more important than positive knowledge in commonsense reasoning including believability assessing and safe exploration strategies. This kind of knowledge needs more attention during building knowledge graphs and ontologies.

Capturing negative knowledge opens the way to use formal reasoning in hybrid systems with generative neural networks generating new strategies (akin to human intuition) while reasoners will check the generated solution against a human-verifiable set of constraints, weeding out the worst errors and paving the way to building trustworthy AI applications.

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
