# OpenReview forum: "Shortcomings of Modern Large-Scale Common Sense Knowledge Graphs"
_eswc-conferences.org/ESWC/2021/Conference/Poster_and_Demo_Track — Submitted to ESWC2021 P&D_

### Official Review · AnonReviewer2 · 2021-04-08
**The paper discusses well-known problems in AI and reasoning, but it is not clear the contribution of the author**

**Rating:** 4
**Confidence:** 5

**Review:**

The paper describes some limits of current common-sense knowledge graphs. However, the discussion is focused on the open-world assumption and the lack of negative knowledge.
These are well-known problems in knowledge/ontology building, and the contribution provided by the author is not novel and clear.
Finally, the author states that negative knowledge could be used to check/verify knowledge automatically generated by neural network/probabilistic approaches.

The paper discusses well-known problems in AI and reasoning, but it is not clear the contribution of the author. In other words, I did not gain new knowledge by reading this paper.

**Anonymity:**

Yes, I would like my review to remain anonymous.

---

### Official Review · AnonReviewer1 · 2021-04-12
**opinion piece about the limitations of current commonsense knowledge graphs, grounded in examples**

**Rating:** 6
**Confidence:** 3

**Review:**

The crux of this paper is that "In artificial intelligence, weeding out wrong options is a good strategy for common sense question answering to multiple-choice questions; it also can be used in natural language processing as natural-language syntax and semantics often rely on believability assessing for disambiguation. However, the negative knowledge required for it is mostly absent in existing common sense knowledge graphs." This seems to me a worthwhile thing to argue. Overall, the paper could be more clear. But for a poster/position piece I think it is sufficiently worthwhile.

COMMENTS
Page 2: an example of a "glaring error" (in addition to the citation) would be useful. (The cited paper talks about misclassification due to a small perturbation - which affects the AI but not the human; this is a glaring error - but it's not so clear to me how this relates to commonsense reasoning.)

Page 2: explain what you mean by '"harmonic" solutions'

Page 2: It is not clear whether you are critiquing open world (as a thing in itself) vs. open world (as a limitation of existing knowledge graphs). I think it is the latter - but you could make this more clear.

Page 3: "we get an unsystematic list of things at least one dog is capable of that is not quite useful in formal reasoning for most tasks." This is a good point. You could bring it even more to the fore, perhaps with an example of how the human reasoning would proceed compared to how the knowledge-graph based automatic reasoning would proceed.

Page 4: This is a very important point: "Negative knowledge is so implicit to our thinking that we rarely feel the need to write it down, much less to formalize it. So it is difficult to acquire by text mining." The implications you draw from it (about negation and "only") are useful.

Minor issues:
page 1: "using bottom-up approach" -> "using a bottom-up approach"
page 2 onwards: right curly quotes at the beginning of a quotation. Use `` in LaTeX to get left quotes
page 3: "pulling a sleigh of smelling drugs" Do you mean "or" here?
page 4: "should provide distinction between" -> "...a distinction..."
page 4: "a variation of universal quantifier" -> "a variation of the universal quantifier"
page 5: "fish does not fly" -> "a fish does not fly" or (plural) "fish do not fly"
page 5: "most of the attention was spent " -> "...has been spent"

**Anonymity:**

Yes, I would like my review to remain anonymous.

---

### Official Review · AnonReviewer3 · 2021-04-16
**The viewpoint of this paper is interesting and reasonable, but the proposed method is weak.**

**Rating:** 4
**Confidence:** 3

**Review:**

This paper points out limitations of exiting Common Sense Knowledge Graphs such as ConceptNet, WebChild, ATOMIC, and COMET.
The main claims are followings.
- Most of their usages are restricted to information search and retrieval.
- Existing methods to use them give probabilistic, instable results.

The author considers that the reasons of the above limitations are ...
- They are under Open World Assumption.
- They lacks negative knowledge.

On the based of these considerations, the author proposes to introduce more negative knowledge to Common Sense Knowledge Graphs with some methods.
I think that these considerations are reasonable while the proposed method to collect negative knowledge is not so much novel.
it is because they are only basic idea at the present.
I suppose that the authors should consider more concreate method with some practical trials.




**Anonymity:**

Yes, I would like my review to remain anonymous.

---

### Decision · Program_Chairs · 2021-04-19

Reject